# Study on Bond Performance between Corroded Deformed Steel Bar and DS-ECC

**DOI:** 10.3390/ma15249009

**Published:** 2022-12-16

**Authors:** Tongwei Liu, Xinping Li, Jialing Che

**Affiliations:** 1School of Civil and Hydraulic Engineering, Ningxia University, Yinchuan 750021, China; 2Ordos Company of Inner Mongolia Tobacco Company, Ordos 017000, China; 3Ningxia Hui Autonomous Region Civil Engineering Disaster Prevention Engineering Technology Research Center, Yinchuan 750021, China

**Keywords:** ECC, desert sand, corrosion rate, bond strength, bond toughness

## Abstract

In order to study the bond performance between desert sands engineered cementitious composites (DS-ECC) and corrosion steel bars, seven groups of specimens were designed and manufactured. Through the center pull-out test, the effects of different types of desert sands, the rate of corrosion (0, 5, 10 and 15%), and the anchorage length of steel bars (5d and 8d) on the bonding properties of DS-ECC and corrosion steel bars were studied. Moreover, a de-rusting agent was used to remove the corrosion, and three groups of specimens were pulled out from the center of the de-rusted steel bars. The results showed that both Tengger DS-ECC and Mu Us DS-ECC have good bond properties with corrosion steel bars. The bond stress slip curves between DS-ECC and corrosion steel bars can be divided into four stages: the micro-slip, slip stage, failure stage and residual stage. The bond stress slip curves between DS-ECC and de-rusted steel bars can be divided into the micro-slip stage, failure stage and residual stage, and splitting and pulling-out failure occurs in DS-ECC specimens. The ultimate bond strength is the highest when the corrosion rate is 5%. The bond toughness index is positively correlated with the anchorage length of steel bars, and negatively correlated with the corrosion rate of steel bars. According to the test results, the bond–slip mathematical relationship is established.

## 1. Introduction

Engineered Cementitious Composites (ECC) are a high-performance material [1,2,3] with strain-hardening and multi-cracking properties based on the micromechanical design theory [4] proposed by Professor Victor C. Li of the University of Michigan, U.S.A. [5,6]. Its ultimate tensile strain is excellent, capable of reaching 3–12%, 150–1200 times that of ordinary concrete [7,8]. Using ECC to strengthen damaged structures can effectively improve their bearing capacity and ductility [9,10]. Therefore, to promote the application of ECC in steel bar projects, it is important to study the bonding properties of ECC with steel bars.

When a reinforced concrete structure is subjected to a complex external environment, its internal steel bars are susceptible to corrosion failure [11]. The corrosion of steel bars reduces the effective cross-sectional area of steel bars [12,13], the tensile strength and ductility of steel bars [14,15,16] and the bond strength between steel bars and concrete [17,18], limiting the load-bearing capacity and durability of reinforced concrete structures [19,20,21,22]. To ensure the safety and service life of the structure, the use of ECC to reinforce steel bar concrete structures with bonding properties between the two is critical, ensuring the success of the steel bars. Lee [23] and Choi [24] investigated the influence of the embedded length of steel bars on the bond performance between ECC and steel bars. Cai et al. [25] through a comparative study of 12 groups of specimens, found that the bond strength between deformed steel bars and ECC is significantly higher than that between deformed steel bars and concrete. Deng et al. [26], through the direct pull-out test, analyzed the influence of steel bar diameter, shape, ECC thickness, strength and fiber content on the bond performance between steel bars and ECC. According to the distribution of bond stress along the embedded length, an accurate bond–slip relationship was established. Hou et al. [27,28,29] conducted an experimental study on the bond strength of corrosion steel bars and an ultra-high toughness cementitious composite (UHTCC). The test results show that the bond strength first increased and then decreased with the increase in corrosion rate, and gradually decreased with the increase in bond length. In sum, the research on the bonding properties of ECC and steel bars is improving, but ECC with quartz sand as fine aggregate is expensive and has a high carbon footprint, while desert sand is abundant and inexpensive. Using desert sand instead of quartz sand as fine aggregate for ECC to make a desert sand engineered cementitious composite (DS-ECC) and applying this to reinforce buildings can save energy, reduce the cost of construction and achieve high-value use of desert sand resources. This research group used desert sand from the Mu Us and Tengger Deserts to replace river sand and prepare DS-ECC, and carried out mechanical and durability tests [30,31,32], but the research on the adhesive properties of DS-ECC and corrosion steel bars is still insufficient.

In view of the lack of current research, this study will further analyze the bond performance of corrosion steel bars and DS-ECC, discuss the influence of desert sand type on the bond performance of DS-ECC and corrosion steel bars, analyze the influence of different steel bar corrosion rates on the bond performance of DS-ECC and corrosion steel bars and further explore the influence of steel bar corrosion rates on the bond performance of DS-ECC and de-rusted steel bars through de-rusted corrosion steel bars. Then, the energy ratio method will be used to calculate the bond toughness index, and variations in the rules of steel bar corrosion rate, anchorage length and bond toughness index will be obtained. Finally, the bond–slip mathematical relationship between corrosion steel bars and de-rusted steel bars and DS-ECC will be established to provide some theoretical basis for the study of the bond performance of DS-ECC and corrosion steel bars.

## 2. Materials and Methods

### 2.1. Test Materials and Their Mechanical Properties

DS-ECC consists of ordinary silicate cement, fly ash, steel bars, PVA fiber made in China, fine sand (Tengger Desert sand and Mu Us Desert sand), water and high-performance water-reducing agent. The cement is P.O 42.5 cement produced by Saimaa Cement Factory. The fly ash is Class F fly ash produced by Ningxia Dam Power Plant, with a specific surface area of 1110 m^2^/kg, a water content of 0.2%, burn loss of 2.8%, free calcium oxide content of 0.8% and sulfur trioxide content of 1.5%. The steel bars are HRB400-grade steel bar produced by Ningxia Shenyin Special Steel Co. The water-reducing agent is a high-efficiency water reducing agent of a polycarboxylic acid, with water reducing efficiency of up to 30%. KS-7003 de-rusting agent, produced by a company in Tianjin, is used as the de-rusting agent. The PVA fiber volume admixture was 2%. The mixing water was tap water. The physical properties of Tengri and Mu Us Desert sand are shown in Table 1. The particle size distribution of desert sand is shown in Figure 1. The fiber parameters are shown in Table 2, where T indicates Tengger Desert sand ECC, M indicates Mu Us Desert sand ECC. The mechanical properties of DS-ECC are shown in Table 3, and the mechanical properties of reinforcing steel are shown in Table 4.

### 2.2. Specimen Design

In this test, 10 groups of DS-ECC with corrosion steel bars center pullout specimens were designed, along with desert sand type, steel bars corrosion rate, anchorage length of steel bars, and whether the steel bars were de-rusted or not as variables. Three specimens were made for each group, and the main design parameters of the specimens are shown in Table 5.

The central drawing specimen was made by a cube wood glue board mold with 150 mm side length, and a circular hole was reserved in the center of both ends of the mold for the arrangement of 12 mm diameter HRB400 steel bars with 5 d and 8 d bonded anchorage lengths, respectively. The specific dimensions of the specimen are shown in Figure 2. When making the specimens, epoxy resin gel was used to wrap the non-corrosion area of the steel bars to reduce the test error. The PVC pipe can control the bonded anchorage length and eliminate the error caused by the concentrated stress on the test results. The gap between the PVC pipe and the steel bars was filled with glass glue to prevent the DS-ECC slurry from leaking into the PVC pipe to ensure the length of the bonded anchorage area.

### 2.3. Determination of Corrosion and Corrosion Rate of Steel Bars

#### 2.3.1. Electrochemical Corrosion

NaCl solution was prepared with a concentration of 3.5%. The drawing specimens maintained to the specified age were placed into the NaCl solution; the liquid surface did not touch the steel bars. Three specimens in each group were connected in series. The free end of the steel bars was connected with the positive pole of the constant voltage and constant current source, the stainless steel piece was connected with the negative pole of the constant voltage and constant current source and the current density and the energizing time were adjusted according to Faraday’s law. The current density was controlled at 200 μA/cm^2^, and the energizing time was calculated by Equation (1):(1)$$t=\frac{zF\Delta m}{MI}$$
where: Δm is the loss of steel bars quality; F=96500 C/mol is Faraday’s constant; z=2 is the number of iron ion charges; M=56 g/mol is the molar mass of iron; and I is the corrosion current intensity

#### 2.3.2. Determination of Steel Bars Corrosion Rate

After the end of the drawing test, the internal steel bars were removed from the split specimen. According to the relevant provisions in the test method for the long-term and durability of ordinary concrete (GB/T 50082-2009), the concrete debris adhering to the surface of the steel bars were removed, the bonded section of the steel bars were intercepted, and the steel bars were pickled with 12% hydrochloric acid solution. To make the pickling uniform, attention was paid to turning over the steel bars during pickling. After the completion of the pickling, the steel bars were soaked and neutralized in clarified lime water, then washed with clean water, and dried in the dry container for 4h with electronic balance (accurate to 0.001 g). The weighing results were used to calculate the actual corrosion rate of the steel bars.

#### 2.3.3. Steel Bars De-Rusting

Through the process described in Section 2.3.1, the steel bars were rusted to the target corrosion rate (5, 10 and 15%), and KS-7003 de-rusting agent was used to remove the rust of the corrosion steel bars. In the process of rust removal, the corrosion steel bar were completely immersed in the de-rusting agent. However, this rust removal method produces stimulating gas and a lot of heat, so it was carried out in a safe, ventilated area. After the corrosion products on the steel bar surface were dissolved in the rust remover, the steel bar was taken out, and the residual rust remover on the steel bar surface was rinsed with clean water. After drying the water on the steel bar surface, the weight was recorded.

### 2.4. Specimen Preparation and Maintenance

The raw materials were weighed according to the test mix ratio, and the desert sand, cement and fly ash were successively poured into the mixing pot for mixing for 1 min; then, water and the water-reducing agent were added for 2 min and, finally, PVA fiber was added for 2 min. After mixing was completed, the mixture was divided into two molds and vibrated on the shaking table to form. The specimen was demolded after 24 h of indoor molding and immediately moved to the standard curing room (temperature 20 ± 2, relative humidity ≥ 95%) to cure for 28 d.

### 2.5. Loading Device and Loading System

The center pull-out test was conducted on a 100 t electro-hydraulic servo universal testing machine (SHT4106) with the test setup shown in Figure 3. Two displacement gauges were installed at the loaded and free ends of the steel bars to measure the relative slip of the steel bars and the specimen at the loaded and free ends, respectively. The loading rate was 1 mm/min, and the test was stopped after the steel bars were pulled out to reach the stable load or the steel bars yielded.

## 3. Results and Discussion

### 3.1. Specimen Failure Mode

The failure modes of the central pullout specimens were all splitting and pulling-out failure, and the failure pattern is shown in Figure 4. In the initial loading stage, there was no crack on the surface of the specimen, and the loading end appeared to slip, while the free end did not slip. When the load was loaded to a certain value, the surface of the specimen began to appear in small cracks. When the loading continued, the surface crack of the specimen extended further, but no penetrating crack was formed, and the free end began to slip. When loading to the maximum load, the slip of the steel bar continued to increase and the specimen is destroyed. The sounds of the DS-ECC matrix splitting and fiber tearing accompanied the loading process, and the specimens maintained good integrity before and after loading.

### 3.2. Bond Stress–Slip Curve Analysis

The average bond–slip curves of each group of specimens are shown in Figure 5. From the figure, it can be seen that as the corrosion rate of steel bars increases, the descending section of the bond–slip curve of corrosion steel bars and DS-ESS is steeper and the strength of the residual section is smaller, while the falling section of the bond–slip curve of de-rusted steel bars and DS-ECC is smoother and the residual section strength is smaller.

The typical bond stress–slip curve (T) between the corrosion steel bars and DS-ECC and the typical bond–slip curve (RT) between the de-rusted steel bars and DS-ECC are shown in Figure 6. The curve T can be divided into four stages: micro-slip stage (OA), slip stage (AB), failure stage (BC) and residual stage (CD). The curve RT can be divided into three stages: micro-slip stage (OE), failure stage (EF) and residual stage (FG).

(1)Micro-slip stage (OA, OE): the chemical bonding force between the steel bars and DS-ECC gradually decreases, the steel bars start to slip, and the bond–slip curve shows a linear increasing trend with the increase of the slip in the steel bars. The curve T rises to point A and the curve RT rises to point E.(2)Slip stage (AB): With the increase in stress, the DS-ECC matrix produces radial cracks along the direction of steel bars’ distribution, at which time the bridging effect of fibers starts to consume energy to retard the development of cracks. With continued loading, the steel bars and DS-ECC bond interface are extruded and sheared, the bond stress reaches the peak stress and the slip phase ends. The effective height of the cross-rib of the de-rusted steel bars is reduced, which leads to a reduction in the bond area between DS-ECC and steel bars and reduces the fiber-bridging effect, so the curve RT has no slip phase.(3)Failure stage (BC, EF): The bond stress is mainly provided by the mechanical bite force and friction force of the inter-rib DS-ECC and matrix DS-ECC, and the bond stress gradually decreases with the increase in the slip value of the steel bar. As the bond interface between the steel bars and DS-ECC is gradually smoothed, the mechanical biting force of the DS-ECC matrix and the DS-ECC matrix between the ribs of the steel bars decreases, and the decreasing range of bond stress slows down.(4)Residual stage (CD, FG): The pull-out interface between the steel bars and the substrate tends to level off, the mechanical bite between the DS-ECC and DS-ECC substrate between the cross ribs of the steel bars is consumed and the residual bond stress changes less, at which time the bond stress mainly depends on the sliding friction between the steel bars and DS-ECC.

### 3.3. Effect of Desert Sand Type on Bonding Performance

The relationship between the two DS-ECC and the bonding properties of corrosion steel bars is shown in Figure 7. Tengger DS-ECC and Mu Us DS-ECC have good bond properties with corrosion steel bars. When the steel bars’ anchorage length is 5 d, the bond strength of T-R10-5 specimen is 8% higher than that of M-R10-5 specimen. When the steel bars’ anchorage length is 8d, the bond strength of T-R10-8 specimen is 9% higher than that of M-R10-8 specimen. It can be seen that the bond strength between Tengger Desert sand ECC and corrosion steel bars is higher than that between Mu Us Desert sand ECC and corrosion steel bars when the anchorage lengths are 5d and 8d. This is because the average particle size of sand in the Tengger desert is lower than that of sand in the Mu Us Desert, and the DS-ECC slurry prepared with Tengger Desert sand has greater uniformity and more compactness, which makes the compressive strength of ECC sand in the Tengger Desert higher than that of ECC sand in the Mu Us Desert. The increase in compressive strength improves the shear performance and mechanical bite force of DS-ECC and steel bars. The bonding strength between DS-ECC and corrosion steel bars are increased. This is in line with the development rules of the bonding properties of ECC and deformed steel bars with compressive strengths of 51.7MPa, 71.0MPa and 81.3MPa studied by Deng et al. [33], and the compressive strengths studied by Liu et al. [34] of 39.4MPa, 48.7MPa and 62.7, respectively. The bonding properties of light aggregate concrete (MPa) and steel bars (83.2MPa) are the same.

### 3.4. Effect of Corrosion Rate of Steel Bars on Bonding Performance

Figure 8 shows a comparison diagram of the ultimate bond stress of corrosion steel bars and Tengger Desert sand ECC under different corrosion rates. It can be seen from the figure that, with the increase in the steel bars’ corrosion rate, the ultimate bond stress between the steel bars and DS-ECC presents a trend of first increasing and then decreasing. When the steel bars corrosion rate is 5%, the ultimate bond stress of the test piece is 6.5% higher than that of the non-corrosion test piece, at which time the bond performance between the corrosion steel bars and DS-ECC is the best; because the slight corrosion increased the surface roughness of the steel bars, the corrosion products filled the micro-cracks between the steel bars and the DS-ECC matrix and the friction between the DS-ECC matrix and the steel bars increased, as did that between the DS-ECC matrix and the steel bar grip force, indicating that the steel bar corrosion rate is small. The corrosion of steel bars will increase the ultimate bond stress between the steel bars and DS-ECC. When the corrosion rate of steel bars increases from 5% to 10% and 15%, the ultimate bond stress of the specimen decreases by 8.7% and 20.4%, respectively. On the one hand, with the increase in the corrosion rate of steel bars, the corrosion products of steel bars are dissolved in the permeate water and leached through the corrosion pit, reducing the grip of DS-ECC matrix on steel bars. On the other hand, the mechanical bite force between steel bars and DS-ECC matrix is reduced due to the damage of the transverse rib of steel bars under a high corrosion rate [27]. The bond strength and ultimate bond stress are reduced, indicating that an excessive corrosion rate will reduce the bond strength between steel bars and DS-ECC. This is the same as the changing trend of bond performance between corrosion steel bars and ultra-high toughness cementitious composite (UHTCC) with different corrosion rates, as studied by Hou et al. [27], and the changing trend of bond performance between longitudinal steel bars and stirrup with different corrosion rates and concrete, studied by Zheng et al. [35].

### 3.5. The Effect of De-Rust on Bonding Performance

Figure 9 shows a comparison of the bond strength of corrosion steel bars and de-rusted steel bars and DS-ECC. It can be seen from the figure that the bond strength between de-rusted steel bars and DS-ECC is 81%, 68% and 46% of the bond strength between non-corrosion steel bars and DS-ECC, respectively, and 80%, 70% and 54% of the bond strength between de-rusted steel bars and DS-ECC when the theoretical corrosion rate of steel bars is 5%, 10% and 15%, respectively. In summary, the bond strength between de-rusted steel bars and DS-ECC decreases with the increase in the corrosion rate of steel bars before de-rusting. This is because the volume expansion of steel bar corrosion products improves the binding force of the matrix DS-ECC to the steel bars. During loading, the corrosion products are compressed and compacted to provide the mechanical bite force and the transverse ribs of the steel bars. The height of the transverse ribs of the de-rusted steel bars is reduced. The relative bonding area between the DS-ECC matrix and the steel bars is small [36]. The mechanical bite force between the steel bars and the matrix DS-ECC significantly decreases. At the same time, the bonding stress is mainly provided by the mechanical bite force and the chemical bonding force; therefore, the bond strength between the corrosion steel bars and DS-ECC is higher than that between the de-rusted steel bars and DS-ECC.

### 3.6. Bond Toughness Index

This study combines the characteristics of the average bond–slip curve of DS-ECC and corrosion steel bars, and refers to the American Society for Testing and Materials (ASTM) C1018 using the energy ratio method to calculate the toughness index. The proposed bond toughness index calculation method that is applicable to DS-ECC and corrosion deformed steel bars uses following equation.
(2)$$I_{1.0}=\frac{S_{\mathrm{OABE}}}{S_{\mathrm{OAD}}}$$
(3)$$I_{0.75}=\frac{S_{\mathrm{OACF}}}{S_{\mathrm{OAD}}}$$
where: $I_{1.0}$ and $I_{0.75}$ are the bond toughness indices corresponding to the bond stress to the end of the slip section and the bond strength at 75%, respectively; $S_{\mathrm{OAD}}$ is the area under the corresponding curve at the beginning of the fiber bridging action (see Figure 10), kN·mm; $S_{\mathrm{OABE}}$ and $S_{\mathrm{OACF}}$ are the areas under the corresponding curves at the end of the slip section and the bond strength at 75%, respectively, kN·mm; A is the end of the micro-slip section (the first inversion point of the rising section); B is the point of peak bond strength; and C is the point of 75% peak bond strength.

The calculation results of the bond toughness index are shown in Table 6. From Table 6, it can be seen that, compared with T-R10-5, the bond toughness index $I_{1.0}$ of T-R10-8 increases by 140%, the bond toughness index $I_{0.75}$ increases by 80% and the bond toughness index $I_{1.0}$ of M-R10-8 increases by 90% compared with M-R10-5. The bond toughness index $I_{0.75}$ increased by 9%; that is, the bond toughness indexes $I_{1.0}$ and $I_{0.75}$ were positively correlated with the anchorage length of steel bars. When the anchoring length is 5d, the corrosion rate is 10% and 15% respectively. Compared with the corrosion rate of 5%, the bond toughness index $I_{1.0}$ of the corrosion steel Tengger Desert sand ECC decreases by 54% and 62% respectively, and the bond toughness index $I_{0.75}$ decreases by 58% and 63% respectively; that is, the bond toughness indexes $I_{1.0}$ and $I_{0.75}$ were negatively correlated with the corrosion rate of the steel bars. This shows that as the corrosion rate of the steel bars increases, the toughening and crack-arresting effects of the fiber become less obvious, and the ductility of the bonding properties of the specimen after peak loading worsens.

### 3.7. Bond–Slip Curve Fitting

Based on the existing bond–slip constitutive relation, using polynomial fitting the bond–slip constitutive relationship between steel bars and concrete form is simple. Therefore, this study adopted the polynomial (4) fitting analysis of the existing test data, obtained from the corrosion steel bars and de-rusted steel bars with DS-ECC bond–slip fitting curve, as shown in Figure 11.
(4)$$\bar{\tau }=a+bs+cs^{2}+ds^{3}+es^{4}$$
where: $\bar{\tau }$ is the bonding stress, MPa, and s is the slip, mm. See Table 7 for other parameters.

According to Figure 11a, the fitted curve has a high degree of similarity with the actual curve in the rising section, a slight deviation from the actual curve at the peak, and a slightly larger deviation from the actual curve in the falling section, which is because the specimen undergoes shear-extraction failure. When the specimen reaches the ultimate bond stress, the bond stress of the specimen slowly decreases with the increase in the slip amount, meaning that a large amount of the peak slip collected by the instrument corresponds to the bond stress. Figure 11b shows that the fitted curve has a high degree of similarity with the actual curve in the rising section and a slightly larger deviation from the actual curve in the peak and falling sections, which is because de-rusting significantly reduces the mechanical bite between the steel bars and DS-ECC, resulting in the bond stress reaching the peak in advance. The test results are in good agreement with the fitted results in the rising, peak and falling sections, so Equation (4) can provide a theoretical basis for the study of the bonding performance of rusted and de-rusted steel bars with DS-ECC.

## 4. Conclusions

The purpose of this research is to study the adhesive properties of corrosion steel bars and DS-ECC. Firstly, seven groups of central pull-out tests were used to analyze the influence of the type and size of desert sand and the corrosion rate (0, 5, 10 and 15%), and the anchorage length of steel bars (5d and 8d) on the adhesion between ribbed corrosion steel bars and DS-ECC. Then, three groups of de-rusted steel bars were used to make center drawn specimens, and the adhesion between de-rusted steel bars and DS-ECC was compared and analyzed. The following conclusions are drawn:(1)The bond–slip curve of DS-ECC and corrosion steel bars can be divided into the micro-slip stage, slip stage, failure stage and residual stage, the bond–slip curve of DS-ECC and de-rusted steel bars can be divided into the micro-slip stage, failure stage and residual stage and the failure types of the specimen are all splitting and pulling-out failure.(2)Tengger DS-ECC and Mu Us DS-ECC have good bond properties with corrosion steel bars. With the increase in the steel bar corrosion rate, the bond strength of the Tengger Desert sand ECC and the corrosion steel bars first increases and then decreases, and the ultimate bond strength is the highest when the corrosion rate is 5%. The bond strength between the de-rusted steel bars and ECC of Tengger Desert sand decreases with the increase in the steel bars’ corrosion rate.(3)When the corrosion rate of the steel bars is 10%, the bond toughness indexes $I_{1.0}$ and $I_{0.75}$ are positively correlated with the anchorage length of steel bars. When the anchorage length is 5d, the bond toughness indexes $I_{1.0}$ and $I_{0.75}$ are negatively correlated with the corrosion rate of the bars, indicating that with the increase in the steel bar corrosion rate, the effect of fiber toughening and crack resistance is less obvious, and the ductility of the bonding properties of the specimen after peak loading worsens.(4)The bond–slip mathematical relationship between corroded steel bars and de-rusted steel bars and DS-ECC is established. The bond–slip curve obtained from the test is in good agreement with the fitting curve. The bond–slip mathematical relationship can provide some theoretical basis for studying the bond performance between corroded and de-rusted steel bars and DS-ECC.

## Figures and Tables

**Figure 1 materials-15-09009-f001:**
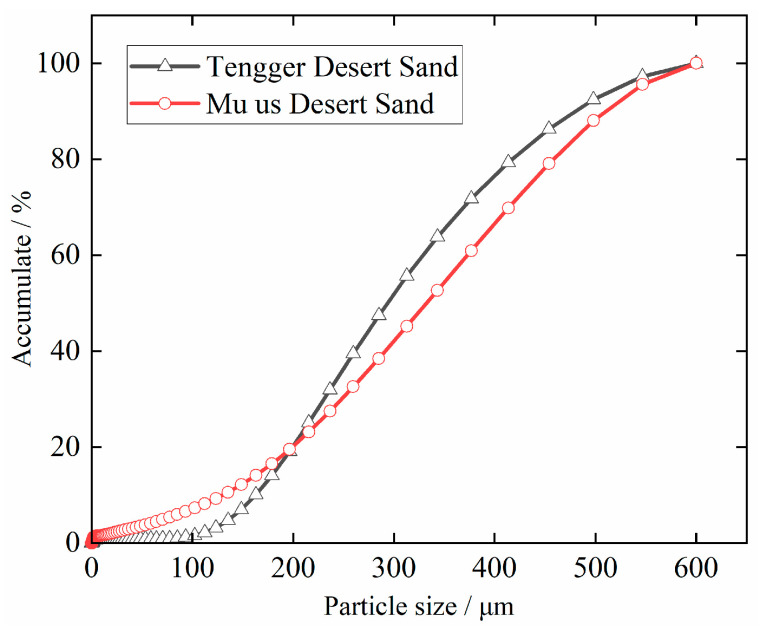
Distribution of desert sand particle size.

**Figure 2 materials-15-09009-f002:**
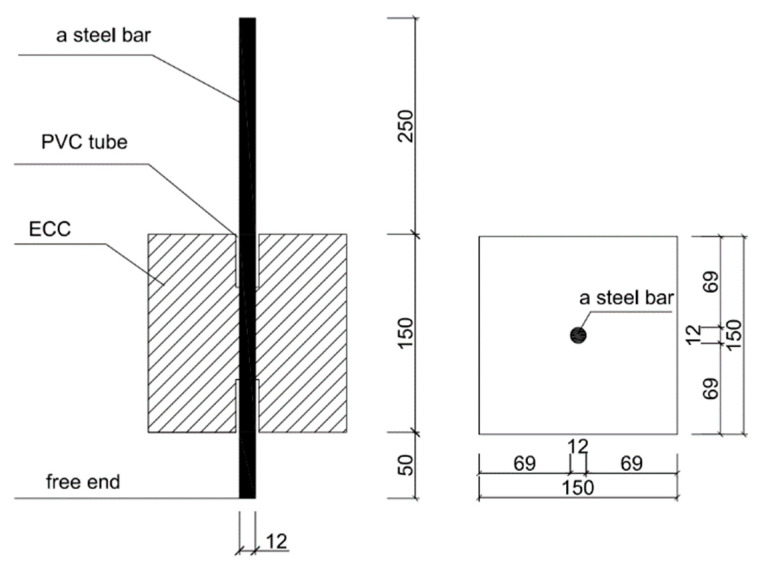
Detailed drawing of the central pullout specimen (unit: mm).

**Figure 3 materials-15-09009-f003:**
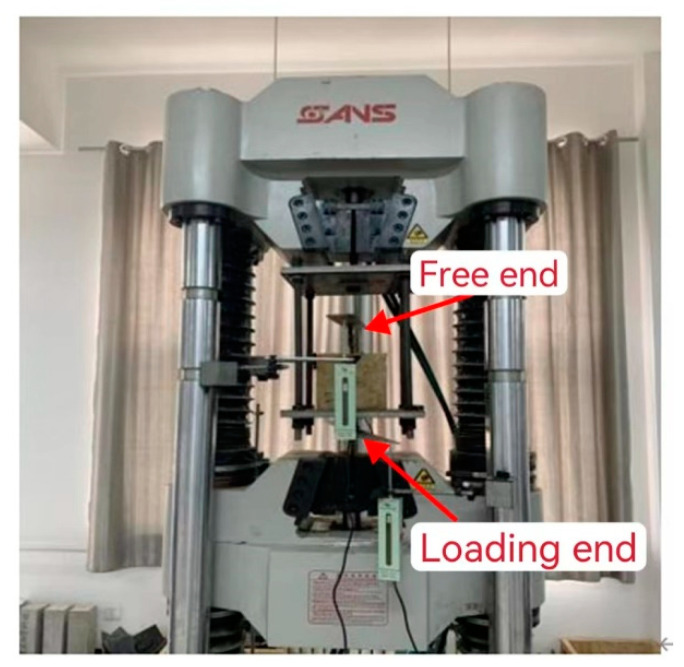
Electro-hydraulic Servo Universal Testing Machine (SHT4106).

**Figure 4 materials-15-09009-f004:**
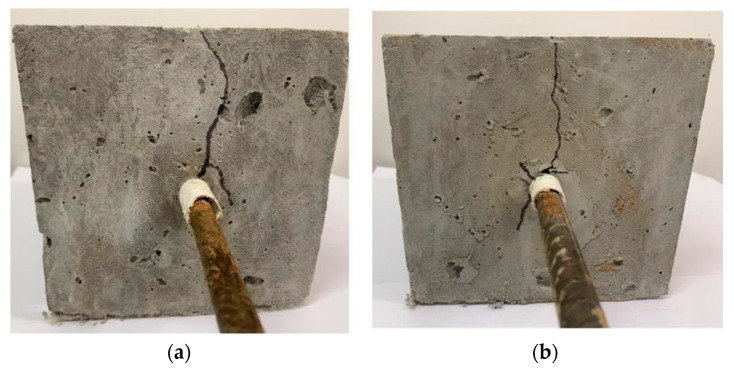
Diagram of splitting and pulling-out failure mode for central pull-out specimens. (**a**) Corrosion steel bar. (**b**) De-rust steel bar.

**Figure 5 materials-15-09009-f005:**
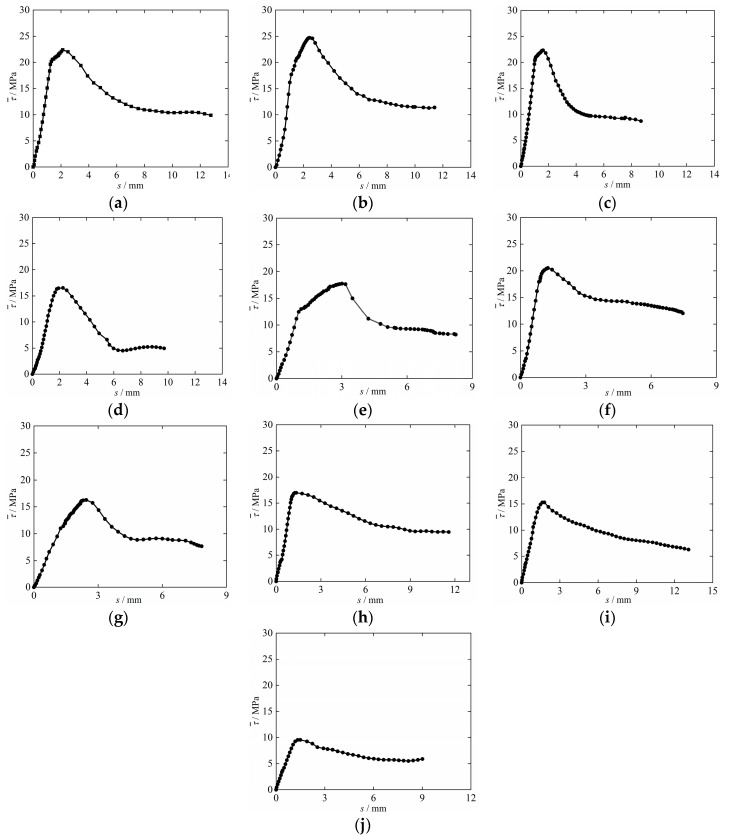
Average bond–slip curve for each group of specimens. (**a**) T-R0-5; (**b**) T-R5-5; (**c**) T-R10-5; (**d**) T-R15-5; (**e**) T-R10-8; (**f**) M-R10-5; (**g**) M-R10-8; (**h**) RT-R5-5; (**i**) RT-R10-5; (**j**) RT-R15-5.

**Figure 6 materials-15-09009-f006:**
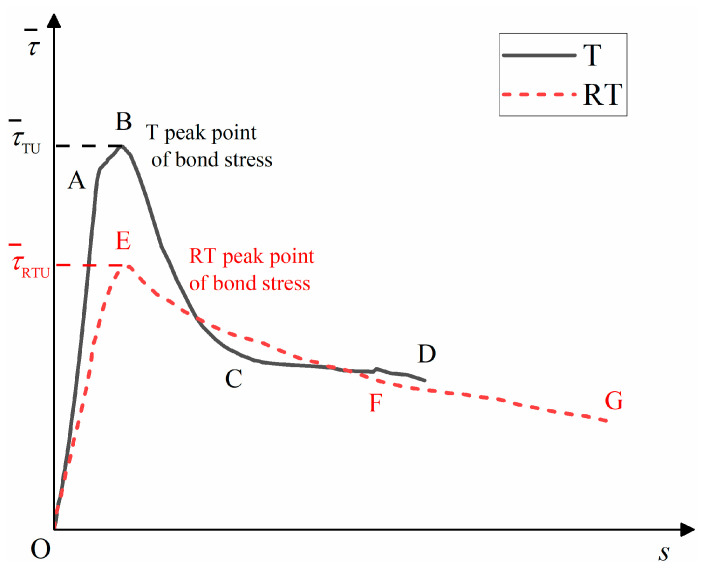
Typical bond–slip curve.

**Figure 7 materials-15-09009-f007:**
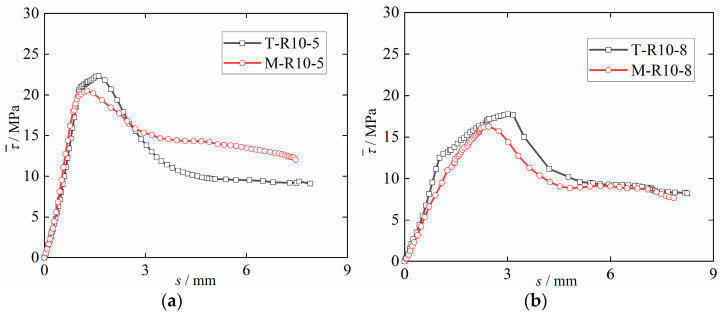
Effect of desert sand type on bonding performance. (**a**) Anchorage length 5d; (**b**) Anchorage length 8d.

**Figure 8 materials-15-09009-f008:**
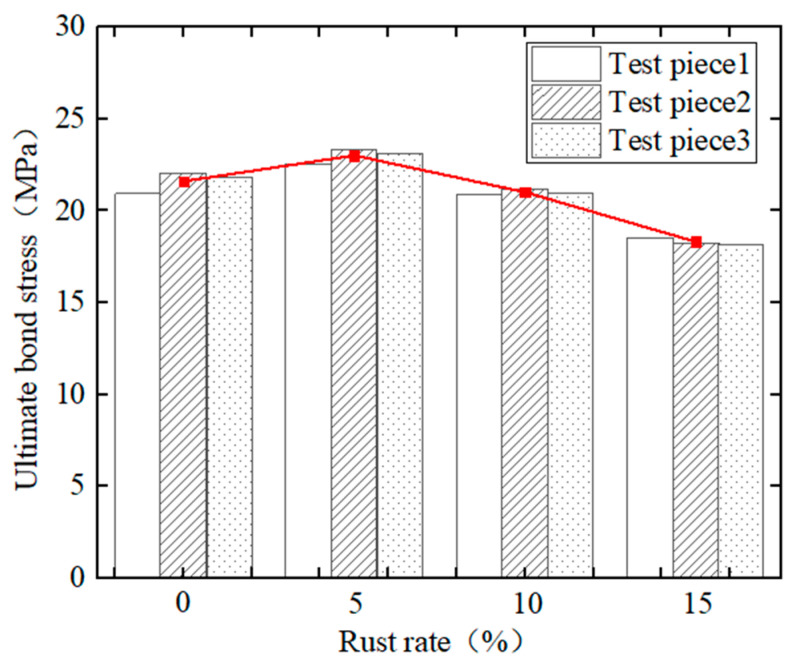
Comparison Diagram of Ultimate Bond Stress of Different Corrosion Rates.

**Figure 9 materials-15-09009-f009:**
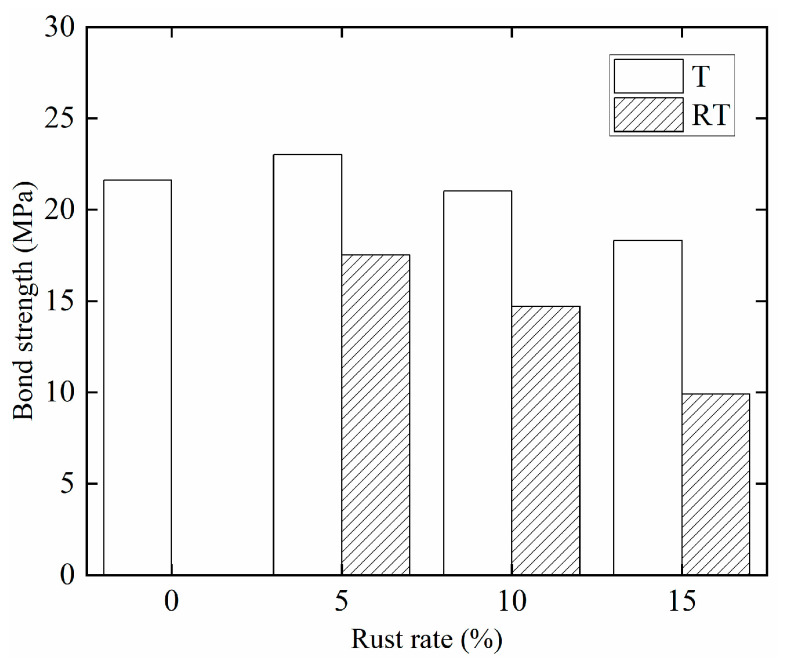
Comparison of bond strength.

**Figure 10 materials-15-09009-f010:**
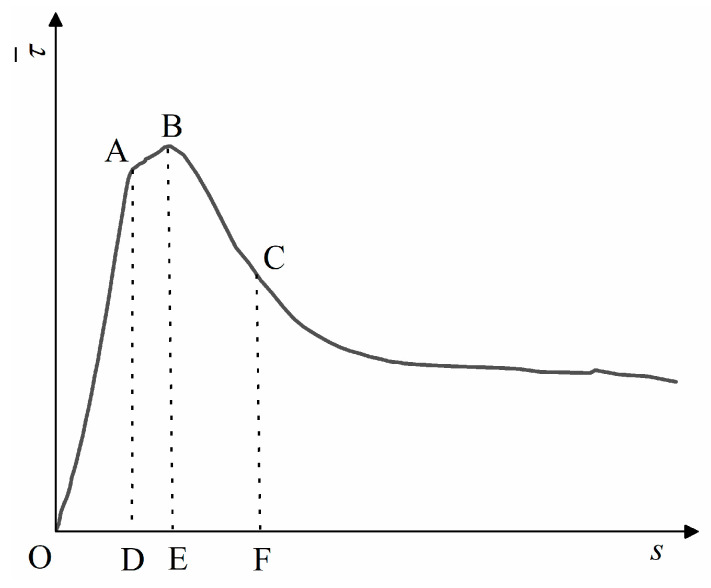
Definitions of toughness indices.

**Figure 11 materials-15-09009-f011:**
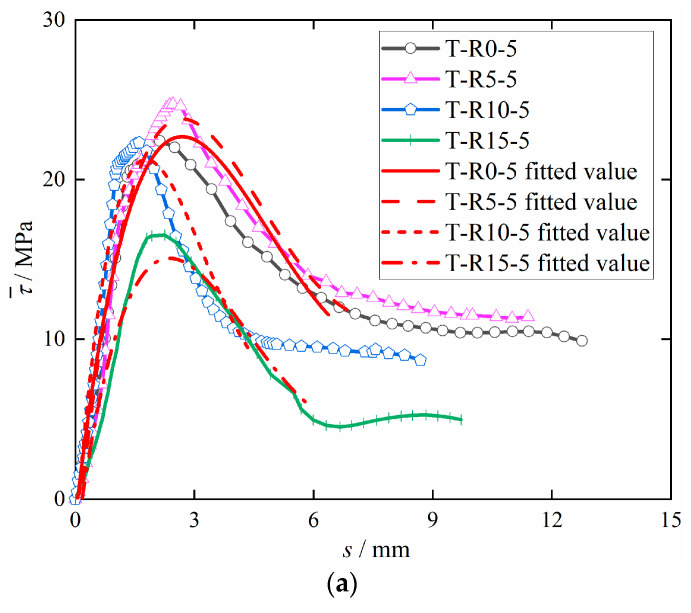
Bond–slip curve test value compared with the fitted value. (**a**) Bond–slip fitting curve of corrosion steel bars with DS-ECC. (**b**) Bond–slip fitting curve of de-rusted steel bars with DS-ECC.

**Table 1 materials-15-09009-t001:** Physical properties of desert sand.

Desert Sand Type	Apparent Density (kg/m^3^)	Bulk Density (kg/m^3^)	Fineness Modulus
Tengger	2623	1562	0.72
Mu Us	2637	1547	0.34

**Table 2 materials-15-09009-t002:** Main parameters of PVA fiber.

Fiber Type	Length (mm)	Diameter (µm)	Ultimate Tensile Strength (MPa)	Density (kL/m^3^)	Elastic Modulus (GPa)
PVA	12	40	1600	1300	40

**Table 3 materials-15-09009-t003:** DS-ECC fitting ratio and mechanical properties.

Concrete Type	Desert Sand Type	Water Binder Ratio	Cement (kg/m^3^)	Fly Ash (kg/m^3^)	Desert Sand (kg/m^3^)	Fiber Content (%)	Average Compressive Strength (MPa)	Tensile Strength (MPa)	Ultimate Tensile Strain (%)
T	Tengger	0.3	617.6	617.6	580.2	2	52.9	9.8	3.3
M	Mu Us						49.8	9.96	4.6

**Table 4 materials-15-09009-t004:** Mechanical properties of deformed steel bars.

A Steel Bar	Diameter (mm)	Average Yield Strength (MPa)	Average Tensile Strength (MPa)	Elongation (%)
HRB400	12	465	625	22

**Table 5 materials-15-09009-t005:** Main design parameters of the test piece.

Number	Anchorage Length (mm)	Theoretical Corrosion Rate (%)	The Actual Corrosion Rate of Steel Bars (%)
T-R0-5	60	0	0
T-R5-5	60	5	5.16
T-R10-5	60	10	9.95
T-R15-5	60	15	14.79
RT-R5-5	60	5	6.32
RT-R10-5	60	10	12.14
RT-R15-5	60	15	15.72
T-R10-8	96	10	10.32
M-R10-5	60	10	9.05
M-R10-8	96	10	8.76

T denotes Tengger Desert Sand ECC. M denotes Mu Us Desert Sand ECC. RT denotes Tengri Desert Sand ECC made with de-rusted steel bars. R0, R5, R10 and R15 denote theoretical corrosion rates of 0, 5, 10 and 15% for steel bars respectively. 5, 8 suffixes denote anchorage lengths of 5d and 8d for steel bars, respectively.

**Table 6 materials-15-09009-t006:** Calculation results of toughness indices.

Number	$S_{\mathrm{OAD}}$	$S_{\mathrm{OABE}}$	$S_{\mathrm{OACF}}$	$I_{1.0}$	$I_{0.75}$
T-R5-5	8.06	37.04	73.81	4.60	9.16
T-R10-5	10.35	21.86	39.47	2.11	3.81
T-R15-5	12.42	21.86	42.16	1.76	3.39
T-R10-8	7.26	36.84	49.47	5.07	6.81
M-R10-5	8.23	14.92	44.95	1.81	5.46
M-R10-8	6.86	23.72	38.15	3.46	5.56

**Table 7 materials-15-09009-t007:** Coefficient in expression (4).

Number	$\bar{\tau }=a+bs+cs^{2}+ds^{3}+es^{4}\;\mathrm{Parameter}$					Relevance R^2^
	a	b	c	d	e	
T-R0-5	−1.217	21.634	−6.333	0.653	−0.022	0.949
T-R5-5	−3.397	25.001	−7.514	0.816	−0.030	0.967
T-R10-5	−2.708	32.657	−14.380	2.222	−0.114	0.923
T-R15-5	−2.947	18.456	−6.111	0.712	−0.028	0.947
RT-R5-5	0.138	17.805	−5.642	0.637	−0.024	0.899
RT-R10-5	0.816	12.186	−3.466	0.351	−0.012	0.894
RT-R15-5	0.172	10.906	−4.165	0.351	−0.012	0.950

## Data Availability

The data presented in this study are available on request from the corresponding author.

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
