# Peer review of "Study on Bond Performance between Corroded Deformed Steel Bar and DS-ECC"

_materials, 2022, doi:10.3390/ma15249009_

Round 1

Reviewer 1 Report

 Ten groups of pull-out samples were tested. The effects of desert sand type, reinforcement corrosion rate, reinforcement rust removal and other factors on the bond performance between DS-ECC and corroded reinforcement were analyzed. 

-          In the abstract, the concrete DS-ECC is named, but it is not explained what it is. It should be done.

-          Page 2. DS-ECC is slightly explained. In that explanation, reviewer does not find differences with a common concrete with PVA fibers. Therefore, what is different in that concrete? It should appear in the text.

-          Related to the previous comment, there are many paper in the scientific literature regarding pull-out test to study the bond between the steel and the concrete. If the concrete is not special, what is the novelty of this research?

-          An extensive English review is needed. For example, the sentence “Install 2 displacement meters at the loaded end of the steel bar, and the difference between the average value of the displacement meters at the loaded end and the elastic elongation of the steel bar at the loaded end is the slip value of the steel bar at the loaded end” is unintelligible.

Reviewer 2 Report

Although the proposed manuscript is interesting, there are enough weaknesses that need to be improved. This based on the following:

·        The abstract should be reviewed again because it is very general and the objective is not clear.

·        The scope of the study is not well defined, the authors could better express it in the abstract

·        The acronyms must be defined: ECC (DS-ECC)

·        The correct meaning of the acronym ECC is “engineered cementitious composites”

·        Authors should correct: ....rate is 0%, 5%, 10% and 15% respectively; should be: ......rate is 0,

·        5, 10 and 15% respectively..

·        This statement is misspelled: Engineered Cement Composite (ECC) is Professor Victor Li of the University of Michigan [13,14] A high-performance material ...

·        The authors should enrich the introduction. In the introduction section, the authors should correct the objective of the study, because it is too long for the readers.

·        This subtitle should be removed: Test Overview and section 2 should be placed, as Materials and methods, as indicated in this Journal.

·        In table 1, 2 y 4, in the subtitles, the slash must be eliminated before the parenthesis begins.

·        In some tables there are words that should reduce the size of the letter and not put a hyphen.

·        All the units in the tables put them in parentheses

·        The units used in Figure 2 must be indicated.

·        Authors should correct: ....rate is 0%, 5%, 10% and 15% respectively; should be: ......rate is 0,

·        5, 10 and 15% respectively.

·        in section 2.3.3 it says .....rate (5%, 10% and 15%)…, the correct thing is ......rate (5, 10 and 15%)…

·        In section 2.4, at the end of the paragraph it says .... for curing to 28d age., you have to eliminate the word "age", it is already implicit that the material has those days of age.

·        The title of figure 3 y 4 are very ambiguous.

·        The authors must indicate which is the section of results and discussions

·        The results only present description of the images and graphics, but there is no discussion of results.

·        It is recommended that the authors review the conclusions and write them in a specific way.

·        The authors present 40 references.  The authors present 6 of their works, which is why 15% is considered self-plagiarism, it must be at least 10% of their works. authors must correct this section

Round 2

Reviewer 1 Report

Authors have addressed all reviewer comments. Therefore the paper is suitable for publication.

Reviewer 2 Report

The authors have made all the suggested corrections, for which the article can be accepted for publication.